# Endophytic Fungi of Olive Tree

**DOI:** 10.3390/microorganisms8091321

**Published:** 2020-08-30

**Authors:** Rosario Nicoletti, Claudio Di Vaio, Chiara Cirillo

**Affiliations:** 1Council for Agricultural Research and Economics, Research Centre for Olive, Fruit and Citrus Crops, 81100 Caserta, Italy; rosario.nicoletti@crea.gov.it; 2Department of Agricultural Sciences, University of Naples Federico II, 80055 Portici, Italy; divaio@unina.it

**Keywords:** *Olea europaea*, endophytes, antagonism, defensive mutualism, plant growth promotion, bioactive compounds

## Abstract

In addition to the general interest connected with investigations on biodiversity in natural contexts, more recently the scientific community has started considering occurrence of endophytic fungi in crops in the awareness of the fundamental role played by these microorganisms on plant growth and protection. Crops such as olive tree, whose management is more and more frequently based on the paradigm of sustainable agriculture, are particularly interested in the perspective of a possible applicative employment, considering that the multi-year crop cycle implies a likely higher impact of these symbiotic interactions. Aspects concerning occurrence and effects of endophytic fungi associated with olive tree (*Olea europaea*) are revised in the present paper.

## 1. Introduction

After evidence resulting from the manifold investigations carried out in the last decades, the awareness that endophytic fungi are constantly associated with plants and remarkably influence their ecological fitness has significantly increased. In fact, the original boost concerning natural ecosystems incited by the general theoretical intent to exploit all components of biodiversity, basically as a source of novel bioactive products, has more recently extended to crops. Within agricultural contexts, the role of the endophytic microbiota, or endosphere, is more consistent in orchards, where the time factor confers higher impact to the establishment of an equilibrium among the species which are part of the tree biocoenosis, as well as to its eventual disruption [1].

The extent at which the accumulating knowledge on the beneficial effects of endophytic microorganisms may have a practical impact in tree crop management, and further progresses can be achieved, is largely dependent on the opportunity by the scientific community and actors in the field to access it in an organized form. In this perspective, the state of the art of research concerning occurrence and effects of endophytic fungi associated with olive tree (*Olea europaea*) are revised in the present paper.

## 2. Relevance of Microorganisms for a Sustainable Management in Olive Growing

The Mediterranean Basin landscape and culture have been shaped by olive tree since ancient times, but the ecological importance of this tree has only recently been acknowledged [2,3]. In the semiarid Mediterranean agricultural lands, new approaches in fruit orchard management have been forced by environmental constrains, such as soil degradation and water shortage, and agronomical techniques that may be able to improve or preserve soil quality and fertility, other than plant health, have gained particular importance [4,5,6,7]. Modern intensification in olive cultivation practices is causing increased incidence and severity of olive pests and diseases; whereas sustainable management systems can positively affect soil biochemical characteristics and soil microbial biodiversity [8,9], and contribute to improve landscape stability, mainly in the rising condition of abandoned olive groves [3]. Thus, year by year a fast-growing percentage of the growers’ incomes is invested in agrochemicals, to promote olive tree growth, to control plant pathogens, and to increase the olive yield and quality, simultaneously generating a great public concern on the negative effects of the agrochemicals use on the environment, on the ecosystem’s biodiversity, and human and animal health [10]. Consequently, several efforts have been done on the development of eco-friendly cultivation practices suitable to sustainable disease control by ameliorating olive tree health and productivity through methods and strategies that promote soil biological processes, decrease agricultural inputs, and improve soil structure and fertility [9].

The diversity of microorganisms associated with plants may stimulate their growth and induce tolerance mechanisms helping plants to counteract adverse environmental conditions. In arid and semiarid environments, crops are facing environmental constraints due to climate-change-driven rising temperatures, changes in rainfall frequency, and occurrence of extreme events [11]. These habitat-elicited stresses may reduce crop productivity and lead to soil erosion and degradation. Plants dwelling in such environments have developed mechanisms helping them to mitigate and counteract abiotic stress. Microorganisms of the rhizosphere can play a pivotal role in health and growth of olive tree too, by establishing strong relationships with the root system that enable plants to grow in limiting conditions, such as water scarcity, salinity, low soil fertility, and so on. In addition to studies on the intrinsic ability of olive tree to adapt to adverse environmental conditions [12], a significant research activity has been performed on rhizosphere microbes providing increased tolerance to host plants under abiotic stress, mainly focusing on plant growth promoting rhizobacteria and arbuscular mycorrhizal fungi [13,14]. Moreover, fungi and actinomycetes have been recognized as able to use root exudates as a carbon source, supplying plants with promptly assimilable nitrates, and playing a crucial role in the maintenance of soil health, besides exerting antagonistic effects on root pathogens [5].

In the plant holobiont system, these beneficial effects are integrated by the microbial component of the endosphere. Endophyte colonization of plants has been recognized to involve a sequence of cross-talking signals that allow the onset of compatible interactions. Once the interrelation has established, endophytes increase stress tolerance through the stress-responsive gene induction/expression, reactive oxygen species and anti-stress metabolite synthesis [15]. Under abiotic stress conditions, endophytic fungi have been reported to produce plant hormones and compatible solutes that maintain integrity and promote growth of the host. Moreover, they are known to protect their host plants against biotic adversities through the production of bioactive compounds and the stimulation of the defense reaction [16]. As soon as interactions between endophytes and plants have been disclosed, it has been argued that they can be exploited for the development of innovative applications in sustainable but still highly productive cultivation systems ([17], and literature therein), similarly to the better known other groups of microbes. As a result of the rising demand for organic agricultural products, perspectives for the application of these microorganisms as potential biopesticides and biofertilizers have become more consistent in the olive sector too [18], along with an incremental interest for the search and identification of species-specific endophytes [19].

## 3. Occurrence and Ecological Implications of Endophytic Fungi of Olive Tree

Table 1 and Table 2 list records concerning occurrence of endophytic fungi in olive tree as inferred from examination of the available literature and GenBank accessions. The first table, dedicated to Ascomycota, is much more numerous; in fact, it includes 245 entries, 116 of which (approx. 47%) are identified at the species level. Such a low proportion can be explained considering that rDNA-ITS sequences are not able to resolve species ascription within many fungal genera [20]. On the other hand, it could reflect the possible existence of novel species, which is quite a common outcome of investigations on endophytic fungi. The same inference could apply to the Basidiomycota series of records, where identification at the species level is occasional, and was achieved for just 7 out of 37 entries (about 19%). Finally, taxa belonging to the Mucoromycota appear to be quite infrequent (Table 2).

With reference to the geographic origin, not surprisingly the great majority of these records come from the Mediterranean region, where olive growing is absolutely dominant in statistical terms with approximately 10.2 million hectares in 2018, corresponding to more than 97% of the overall surface destined to this crop in the world [44]. In addition to geographic and climatic conditions, differences in the species assortment are related to several factors, such as plant organ (Table 1), phenological stage [34], cultivar [22,27,29,32,35], season and cardinal orientation of samplings [23,24,26,29], isolation procedure, and substrate employed [33].

In timeline terms, the great majority of records have been gathered in the last four years, with a significantly increasing trend. In fact, after just three previous investigations carried out in the island of Majorca (Spain) in 1992 [30], Sicily (Italy) in 2008 [23], and Brazil in 2013 [33], and a couple of extemporary findings from Italy [38,41], the available data have been integrated with 48 new records in 2016, 27 in 2017, 68 in 2018, 107 in 2019, and 73 in the first half of the current year. Several reasons can explain such an escalation. Particularly, the increasingly easier access to the molecular tools has remarkably enhanced the number of isolates which can be taxonomically identified, subverting the old classification procedures which basically relied on the investigators’ mycological experience, or on the access to identification services. Previously infrequent or unknown taxa have started being reported thanks to this methodological improvement. However, in many of these records identification was limited at the genus level, particularly when rDNA-ITS sequences only were considered as the genetic marker [21,22,29]. Although acceptable, this reflects a lower significance of the reports, considering that different species within genera such as *Alternaria, Aspergillus, Cladosporium, Diaporthe, Fusarium, Penicillium, Phoma*, etc., may play very different ecological roles. With reference to the identified species, just two common plant associates (*Alternaria alternata* and *Epicoccum nigrum*) have more than two records from different locations, indicating that for the time being no species seems to stand out for a regular endophytic association with olive tree.

Another explanation for this incremental trend is the previously introduced emerging awareness of a relationship between endophyte occurrence and pest and pathogen incidences in crops. In the case of olive tree, this concept has been particularly considered with reference to the outbreak of the quick decline syndrome incited by *Xylella fastidiosa* in southern Italy [45], although no significant associations of any identified endophytes with this bacterium were found in both a high and a low susceptible cultivar in a dedicated study [21]. In another study concerning relationships with another widespread bacterial pathogen, *Pseudomonas savastanoi* pv. *savastanoi* causing the olive knot disease, endophytic fungi were found to be more abundant in infected plants. However, this remark particularly involved potential pathogens, such as *Alternaria, Cladosporium, Pseudocercospora, Fusarium*, and other Nectriaceae [27].

### 3.1. Endophytic Fungi as Plant Disease Agents

The wilt agent *Verticillium dahliae* is probably the fungal pathogen of olive tree which could be considered to have more strict endophytic implications. Basically, it is considered a hemibiotrophic fungus which colonizes olive trees systemically by spreading through the xylem during a biotrophic phase where it causes no or minimal detrimental effects on plant physiology. Symptoms consisting in chlorotic leaves rolling inward, defoliation, necrosis, and branch desiccation become evident later on [46]. Considering that this more or less enduring latent stage is recognized as a crucial phase of the disease cycle, the recovery of *V. dahliae* from asymptomatic plants is conventionally not referred to a possible merely endophytic status. However, its finding in artificially inoculated plants which became asymptomatic after recovering from infection [47] raises questions on whether its occurrence within olive trees is necessarily related to pathogenicity.

Other known disease agents have been isolated from asymptomatic plants. An inventory of olive tree pathogens compiled in 2014 includes at least 12 species reported in Table 1, namely *A. alternata, Alternaria consortialis* (=*Ulocladium consortiale*), *Arthrinium phaeospermum, Berkeleyomyces basicola* (=*Thielaviopsis basicola*)*, Botrytis cinerea, Colletotrichum acutatum, Dothiorella iberica, Epicoccum nigrum, Fusarium oxysporum, Macrophomina phaseolina, Neocosmospora solani* (=*Fusarium solani*) and *Neofabraea vagabunda* (=*N. alba, Phlyctema vagabunda*) [48]. In addition to the above-mentioned common associates of olive trees, *A. alternata* and *E. nigrum* are also known to develop epiphytically [41], for the other species, most of which are anyway not reported to cause relevant damage to this crop, it is not inferable if their endophytic presence documented in references of Table 1 was eventually preliminary to disease onset.

In addition to the above inventory, more taxa listed in Table 1 were recovered from diseased olive trees, albeit without verifying their involvement in etiology through the Koch’s postulates. This is the case of the new species *Phaeomoniella oleae*, which was originally isolated from black-discolored xylem of wilting branch of a plant infected by *X. fastidiosa* [49], as well as *Conyozyma leucospermi*, *Nigrospora oryzae* and *Biscogniauxia mediterranea*, which were isolated from twig cankers in California [50]. However, the latter species has been very recently reported as the agent of a charcoal disease in Tunisia [51]. Other recent reports concern *Diaporthe ambigua* causing twig cankers in Italy [52] and *Neofabraea kienholzii* causing leaf and shoot lesions in California [53], while *Cytospora pruinosa* recovered from plants showing branch dieback symptoms in Spain was found not to be pathogenic [54]. Notwithstanding, reports concerning *Cytospora* [30,35] must be taken with caution, considering recent assessments of pathogenicity on olive tree by a few *Cytospora* spp., including the novel species *C. olivarum* [55,56].

The endophytic occurrence of more fungi only identified at the genus level should be better evaluated as well. In addition to *Venturia* and *Verticillium* found in Portugal [29,34], this applies to strains in the genera *Neofusicoccum* and *Phaeoacremonium*, with reference to records of *P. aleophilum* and several *Neofusicoccum* spp. as agents of branch dieback and decline of olive trees [50,54,57]. Similar considerations are valid for strains of *Diaporthe* (=*Phomopsis*), which are renowned canker agents of common endophytic occurrence on many woody plants [58], *Diplodia, Pestalotiopsis*, and *Phoma*, including species which can be disease agents of olive tree [48]. Above all, this concept concerns endophytic strains of *Colletotrichum* which are frequently reported in asymptomatic plants of both olive tree [27,28,29,33,59] and other crops, such as citrus [60]. Several *Colletotrichum* spp. are involved in etiology of olive anthracnose with various degrees of virulence and latency [61,62,63], including the new species *C. clavatus* to which the findings of *C. acutatum* as endophyte in olive drupes in Italy are likely to be referable [64]. It goes without saying that a more thorough assessment of their endophytic occurrence in the different cropping contexts is expected to have a remarkable impact on the management of anthracnose.

Unlike the above pathogens, the agent of the Dalmatian disease of olives, *Botryosphaeria dothidea* [65], is missing in the list of endophytic fungi, despite this species is best known as an endophyte of a high number of plants [66]. The fact that its documented occurrence is restricted to diseased drupes is possibly linked to the role as a vector by the cecidomyid *Lasioptera berlesiana*, a parasitoid of the olive fruit fly *Bactrocera oleae* [67], considering that *B. dothidea* is constantly associated with many cecidomyid midges [68].

### 3.2. Endophytic Fungi as Mutualists

Many species found as endophytes of olive tree are known to behave as mutualists in crops, based on their ability to contrast pests and pathogens, and/or to promote plant growth; a brief overview of such properties with reference to species included in Table 1 is proposed in this paragraph.

Strains of *Trichoderma*, mainly reported from Portugal and ascribed to at least four species, have already been experimentally evaluated on olive tree, with reference to both kinds of beneficial effects, particularly for the biocontrol of *V. dahliae* and *N. solani* [18,69,70,71]. The typical soil fungus *Penicillium restrictum* has been reported as an antagonist and mycoparasite of several plant pathogens [72], but other *Penicillium* spp. mentioned in Table 1 are also known as endophytic associates exhibiting antifungal effects [73]. Traditionally known as a biocontrol agent of the grey mold agent *B. cinerea* [74], *Clonostachys rosea* (=*Bionectria ochroleuca*) has also disclosed potential against insects and nematodes [75], likewise *Purpureocillium lilacinum* [76]. Already employed in the formulation of biopesticides, the latter species has been reported for antagonism against *V. dahliae* on eggplant, along with plant growth promoting effects [77]. Again, found in association with *V. dahliae* in southern Italy [41], *Paecilomyces variotii* is known for its antagonistic behavior against plant pathogenic fungi in vitro and in vivo [78], as well as nematodes [79]. Species of *Cosmospora* have been reported as mycoparasites [80], and pathogens of armored scales (Hemiptera, Diaspididae) [81]. Surprisingly, some species of common endophytic occurrence which are basically known as entomopathogens or display a dual biocontrol aptitude against both arthropods and fungi, such as *Beauveria bassiana, Metarhizium anisopliae*, and *Lecanicillium/Akanthomyces* spp. [82,83,84], have not been found as endophytes of olive tree so far.

Based on previous citations as fungal antagonists and producers of bioactive secondary metabolites, other species included in Table 1 may have a role in defensive mutualism [12,15,16,19], such as *Chaetomium globosum* [85], and species of *Paraconiothyrium* [86], *Alternaria* [87], and *Epicoccum* [88]. Strains of the two latter genera were found at a significantly higher rate in asymptomatic leaves in a survey carried out in Portugal considering the key leaf pathogens of olive tree *Venturia oleaginea* and *Pseudocercospora cladosporioides*, which may be indicative of an antagonistic role against the above disease agents [32]. Similar considerations have been advanced for *Chromelosporium carneum* and other taxa in the Pezizales (e.g., *Heydenia* and *Pyronema*); in fact, these fungi were more frequent in plants which did not show symptoms of the olive knot disease [27]. Indeed, their role in suppressing this bacterial disease deserves to be further investigated.

Finally, other endophytic associates of olive tree are more reputed for their plant growth promoting potential depending on improvement of nutrient availability and/or production of plant hormones, such as auxins by *Discosia* sp. [89], and gibberellins by *Phoma herbarum* [90]. Moreover, several yeast species are known for these effects, such as *Aureobasidium pullulans*, which stands out for its widespread occurrence on the olive phylloplane too [41,91].

### 3.3. Endophytic Fungi as Neutral Associates

The ecological role of the many identified endophytic fungi which were not mentioned in the previous sections requires further assessments. In the absence of any circumstantial evidence, such associations are usually defined as neutral. However, it is hard to accept that this approximate inference is valid for such a high number of taxa; rather, it is likely that at least some of them are going to disclose better defined ecological relationships in future. As an example, the ability by a plant to support endophytic development and reproduction of fungi which are pathogenic towards other plant species has been envisaged to possibly represent an ecological adaptation supporting the competitive attitude of the plant host [92]. In this respect the available data indicate that olive can be the host of species such as *Cadophora luteo-olivacea, Curvularia trifolii, Dactylonectria pauciseptata, Didymella macrostoma*, *Drechslera avenae, Eutypa tetragona, Paraphoma chrysanthemicola, Parastagonospora avenae, Pestalotiopsis guepinii, Pleospora herbarum, Stagonosporopsis cucurbitacearum*, and *Chondrostereum purpureum*, known as pathogens of various crops.

For other fungi already known for their endophytic attitude on several plant species, such as *Anthostomella leucospermi*, *Arcopilus aureus*, *Chalastospora gossypii*, *Daldinia concentrica*, *Endoconidioma populi*, *Fimetariella rabenhorstii*, *Nemania aenea*, *Paraphaeosphaeria sporulosa*, *Preussia africana*, and *Sporormiella intermedia*, the occurrence on olive tree may be rather interpreted as reflecting a general ecological adaptation to horizontal spread within the phytocoenoses [93].

Because of their recent taxonomic description, other species listed in Table 1 have no significant references in the literature considering their ecological role yet. This is the case of the Dothideomycetes *Leptosphaerulina saccharicola* [94], *Dendrothyrium variisporum* [95], *Libertasomyces platani* [96], and *Stigmatodiscus enigmaticus*, with the latter representing the founder of the new order Stigmatodiscales [97].

Although lichens are quite common epiphytes of olive trees, the endophytic occurrence of some lichenicolous species is to be remarked with reference to their exclusive finding in the mentioned study concerning the *X. fastidiosa* epidemic in Salento, Italy [21]. More in detail it is about the genera *Absconditella, Biatora, Lecania, Lecanora, Lecidella, Leimonis, Xanthoria, Catillaria, Hyperphyscia, Pyrrhospora, Scoliciosporum* and *Xanthoparmelia* (=*Karoowia*), with the last five only found in the cultivar Leccino which is known to be resistant to this bacterial disease.

Finally, a mention is deserved for some taxa which have been described as opportunistic human pathogens, such as *Cladophialophora*, *Exophiala*, *Fusarium musae, Hormonema*, *Lecythophora*, *Rhinocladiella similis*, and the yeasts *Candida*, *Pichia*, *Wickerhamomyces*, *Cryptococcus*, *Malassezia* and *Rhodotorula mucilaginosa* [98].

## 4. Biochemical Properties and Possible Biotechnological Applications

It is generally accepted that endophytes exert defensive mutualism through biochemical interactions with both the host plant and its pests and pathogens. The ability to release bioactive secondary metabolites and enzymes may ensure direct or indirect antagonistic effects and promote the host’s defense reaction. Based on this paradigm, a huge amount of studies concerning microbial endophytes deal with the characterization of products and enzyme complexes and examine perspectives for their biotechnological exploitation [99,100].

So far, the available information concerning endophytic fungi of olive tree is quite limited, and basically concerns known compounds. The acidic terpenoid arundifungin was found as a product of an unidentified Coelomycetes strain recovered as an olive endophyte in Spain, possessing antifungal properties which derive from inhibition of glucan synthesis, likewise the better known echinocandin [101]. A strain of *Penicillium chrysogenum* was found to produce the bioactive indole alkaloids meleagrin, roquefortine C, and dehydrohistidyltryptophenyl-diketopiperazine [42]. Moreover, from the same location of the Egyptian oasis of Siwa, an isolate of *Penicillium citrinum* was reported to produce the polyketide mycotoxin citrinin, along with the pyrrolidine alkaloids 2-(hept-5-enyl)-3-methyl-4-oxo-6,7,8,8a-tetrahydro-4*H*-pyrrolo[2,1-*b*]-1,3-oxazine, scalusamide A, and perinadine A [36].

Endophytic fungi recovered from olive leaves and identified as *Alternaria* sp., *Chaetomium* sp., *Diaporthe* sp., *Fusarium* sp., *E. nigrum*, and *Nigrospora oryzae* were found to possess antagonistic properties against *C. acutatum* in vitro, with the latter species displaying the most consistent effects. Such effects were at least in part related to the production of volatile organic compounds (VOCs), such as phenylethyl alcohol, pyrazines, amine, and propanoic acid derivatives [31].

In other cases, investigations have been limited to a preliminary stage considering culture filtrates or extracts from the same. This is the case of culture filtrates obtained from endophytic strains of *E. nigrum* and *R. similis* displaying inhibitory effects against *P. savastanoi* pv. *savastanoi* [102,103]. Antibiotic activity against both Gram+ and Gram− bacteria has been reported in vitro for strains of *Penicillium canescens*, *Penicillium commune*, and *A. alternata*, with the latter also active against yeasts (*Candida* spp.) [25]. In the case of *A. alternata*, the antibiotic effect was also induced by the ethyl acetate extracts from both mycelium and the culture broth, while in the case of *P. commune*, bioactivity increased after the addition of an olive leaf extract to the medium.

*Penicillium commune* was again reported for in vitro antagonistic effects against the agent of anthracnose (*C. acutatum*); such effects were increased by the placement of an olive leaf in the test plates, indicating that interactions might be more dramatic in the contact with plant host [43]. The same research group pointed out inhibitory effects on mycelial growth of this pathogen by endophytic strains of *Chondrostereum purpureum*, with evident alterations in hyphal structure [28], and of both *C. acutatum* and *V. dahliae* by *Trichoderma lixii* and *P. lilacinum* [40]. Moreover, endophytic strains of *A. pullulans* applied at the blooming stage or just prior to harvest were respectively able to reduce incidence of the anthracnose agent during its latency and in post-harvest on the shelf [59]. The latter effect is confirmatory of reports referring to other fruit crops that VOCs released by endophytic strains of this yeast effectively inhibit spore germination of several post-harvest pathogens [104,105].

An olive endophytic strain identified as *Daldinia* cf. *concentrica* also showed inhibitory effects against a panel of plant pathogenic fungi and oomycota, which were basically dependent on the production of antimicrobial VOCs. Exposure of dried fruits and grains to these volatiles resulted in their full disinfection preventing the development of molds and suppressed infection by *Aspergillus niger* in peanuts [39]. In greenhouse experiments VOCs produced by this strain showed bionematicidal activity against the second-stage juveniles of *Meloidogyne*
*javanica* (67% reduction in viability). Among these volatiles, 4-heptanone elicited the most consistent effect with 90% reduction in viability, while egg hatching decreased by 87%. Moreover, the application of a volatile mixture to soil inoculated with *M*. *javanica* eggs or juveniles significantly reduced galling index in susceptible tomato plants with no effect on root weight [106].

Finally, an interesting ground of investigation has been envisaged for endophytic fungi of olive tree in view of applicative perspectives in the management of *X. fastidiosa* epidemic spread, based on evidence of either repellent or attractant effects that some species possibly exert against its vector, the meadow spittlebug (*Philaenus spumarius*) [107].

## 5. Conclusions

Research on endophytic fungi is gradually evolving from a basically descriptive stage to the analysis of factors determining the structure of microbiomes, in the perspective that their manipulation may enable to increase plant protection and productivity. In this respect, it has been observed that a better comprehension of the genetic interactions with the host tree and other associated microbes is crucial for the success of practical applications of endophytic fungi in sustainable agriculture [108].

In the case of *O. europaea*, the increasing number of reports concerning endophytic fungi in the past few years confirm that their spatial–temporal distribution in olive trees has been poorly investigated. However, as for other crops, the accumulating data support the evidence of the substantial impact of this microbial component of biodiversity on fitness of olive tree. As a further example, the recalcitrance to sterilization protocols for in vitro propagation reported in a Slovenian study is indicative that at least some of these fungi are intimately associated to olive plants and adapted for a long-term survival and proliferation in their tissues [37].

Increasing knowledge about functions and dynamics of endophytic communities is fundamental in the aim to exploit use as biocontrol agents. The epidemiological relevance of these microorganisms is basically related to a modulatory role in the spread of cryptogamic diseases. Even when there is no apparent direct interaction with disease agents, the possible effect by endophytic fungi in stimulating plant defense reaction, or more in general to act as plant disease modifiers [109], should not be disregarded. In this respect, data concerning occasional isolations might as well disclose some relevance. Indeed, the role of microbial inoculants on protection, growth stimulation, and productivity is now recognized for olive tree too [14], and the expectancy is high that the applicative use of endophytic fungi may soon become an additional tool in the sustainable management of olive growing.

## Figures and Tables

**Table 1 microorganisms-08-01321-t001:** Endophytic Ascomycota reported from *Olea europaea*.

Endophyte ^1^	Plant Part	Country	Reference
*Absconditella* sp.	branch	Salento, Italy	[21]
*Acaulium* sp.	root	Córdoba, Spain	[22]
*Acremonium* sp.	leaf, twig	Sicily, Italy	[23]
	branch, leaf	Salento, Italy	[21]
*Alternaria alternata*	leaf, root	Bragança district, Portugal	[24]
	leaf	Trás-os-Montes, Portugal	[25]
	leaf	Alentejo, Portugal	[26]
	twig	Mirandela, Portugal	[27]
	fruit	Karaburun, Turkey	(GenBank)
*Alternaria arborescens*	leaf	Bragança district, Portugal	[24]
*Alternaria brassicae*	fruit	Mirandela, Portugal	[28]
*Alternaria compacta*	leaf	Alentejo, Portugal	[26]
*Altenaria consortialis*	root	Bragança district, Portugal	[24]
*Alternaria infectoria*	leaf	Alentejo, Portugal	[26]
	twig	Mirandela, Portugal	[27]
*Alternaria murispora*	leaf	Alentejo, Portugal	[26]
*Alternaria preussii*	leaf	Mirandela, Portugal	[29]
*Alternaria solani*	twig	Mirandela, Portugal	[27]
*Alternaria* sp.	stem	Majorca, Spain	[30]
	leaf, twig	Sicily, Italy	[23]
	leaf	Evora, Portugal	[31]
	fruit	Mirandela, Portugal	[28]
	leaf, twig	Mirandela, Portugal	[27,29]
	leaf	Alentejo, Portugal	[32]
	branch, leaf	Salento, Italy	[21]
*Alternaria tenuissima*	stem, xylem	Majorca, Spain	[30]
	twig	Mirandela, Portugal	[27]
*Anthostomella leucospermi*	leaf, twig	Mirandela, Portugal	[27,29]
*Arcopilus aureus*	leaf	Alentejo, Portugal	[26]
*Arthrinium phaeospermum*	stem, xylem	Majorca, Spain	[30]
*Arthrinium* sp.	leaf	Alentejo, Portugal	[32]
	leaf	Salento, Italy	[21]
*Ascochyta* sp.	leaf, twig	Sicily, Italy	[23]
*Ascochytulina deflectens*	stem, xylem	Majorca, Spain	[30]
*Aspergillus* sp.	leaf	Piracicaba, Brazil	[33]
	fruit	Bragança district, Portugal	[34]
	leaf, twig	Mirandela, Portugal	[27,29]
	leaf	Alentejo, Portugal	[26,32]
	root	Córdoba, Spain	[22]
	xylem	Apulia, Italy	[35]
	branch, leaf	Salento, Italy	[21]
*Aspergillus stellatus*	twig	Sicily, Italy	[23]
*Aspergillus tubingensis*	root	Siwa oasis, Egypt	[36]
*Aureobasidium pullulans*	stem, xylem	Majorca, Spain	[30]
	leaf, twig	Sicily, Italy	[23]
	leaf	Alentejo, Portugal	[26]
*Aureobasidium* sp.	leaf, twig	Mirandela, Portugal	[29]
	leaf	Alentejo, Portugal	[32]
	branch, leaf	Salento, Italy	[21]
*Bartalinia* sp.	fruit	Karaburun, Turkey	(GenBank)
*Berkeleyomyces basicola*	root	Bragança district, Portugal	[24]
*Biatora* sp.	branch	Salento, Italy	[21]
*Biscogniauxia mediterranea*	flower buds	Bragança district, Portugal	[34]
	fruit	Mirandela, Portugal	[28]
	leaf	Alentejo, Portugal	[26]
	leaf, twig	Mirandela, Portugal	[27,29]
*Biscogniauxia nummularia*	shoot	Ljubljana, Slovenia	[37]
*Botryosphaeria* sp.	leaf, twig	Sicily, Italy	[23]
*Botrytis cinerea*	leaf, twig	Sicily, Italy	[23]
	leaf	Alentejo, Portugal	[26]
	leaf, twig	Mirandela, Portugal	[27,29]
*Botrytis* sp.	leaf	Alentejo, Portugal	[32]
	branch	Salento, Italy	[21]
*Cadophora luteo-olivacea*	root	Bragança district, Portugal	[24]
*Camarosporium* sp.	leaf, twig	Sicily, Italy	[23]
	leaf, twig	Mirandela, Portugal	[27,29]
*Canalisporium* sp.	root	Córdoba, Spain	[22]
*Candida* sp.	twig	Sicily, Italy	[23]
	branch, leaf	Salento, Italy	[21]
*Capnobotryella* sp.	branch	Salento, Italy	[21]
*Catenulostroma* sp.	branch, leaf	Salento, Italy	[21]
*Catillaria* sp.	branch	Salento, Italy	[21]
*Ceratocystis* sp.	leaf, twig	Sicily, Italy	[23]
*Cercospora* sp.	branch	Salento, Italy	[21]
*Ceuthospora* sp.	xylem	Apulia, Italy	[35]
*Chaetomium globosum*	shoot	Ljubljana, Slovenia	[37]
*Chaetomium* sp.	leaf	Sicily, Italy	[23]
	leaf	Evora, Portugal	[31]
	shoot	Ljubljana, Slovenia	[37]
	leaf	Alentejo, Portugal	[32]
*Chalara* sp.	leaf, twig	Sicily, Italy	[23]
	leaf	Alentejo, Portugal	[32]
*Chalastospora gossypii*	leaf, twig	Mirandela, Portugal	[27,29]
*Chromelosporium carneum*	leaf, twig	Mirandela, Portugal	[27,29]
*Ciboria* sp.	leaf	Salento, Italy	[21]
*Cladophialophora* sp.	root	Córdoba, Spain	[22]
*Cladosporium cladosporioides*	leaf	Alentejo, Portugal	[26]
	fruit	Mirandela, Portugal	[28]
*Cladosporium cucumerinum*	fruit	Mirandela, Portugal	[28]
*Cladosporium delicatulum*	leaf	Alentejo, Portugal	[26]
*Cladosporium herbarum*	leaf	Alentejo, Portugal	[26]
*Cladosporium pseudocladosporioides*	leaf	Alentejo, Portugal	[26]
*Cladosporium ramotenellum*	fruit	Karaburun, Turkey	(GenBank)
*Cladosporium* sp.	leaf, twig	Sicily, Italy	[23]
	shoot	Ljubljana, Slovenia	[37]
	fruit	Mirandela, Portugal	[28]
	leaf, twig	Mirandela, Portugal	[27,29]
	xylem	Apulia, Italy	[35]
	leaf	Salento, Italy	[21]
*Cladosporium sphaerospermum*	shoot	Ljubljana, Slovenia	[37]
*Cladosporium tenellum*	leaf	Alentejo, Portugal	[26]
*Cladosporium tenuissimum*	stem, xylem	Majorca, Spain	[30]
*Clonostachys rosea*	root	Bragança district, Portugal	[24]
*Colletotrichum acutatum*	fruit	Gioia Tauro area, Italy	[38]
	twig	Portugal	(GenBank)
*Colletotrichum nymphaeae*	leaf	Alentejo, Portugal	[26]
*Colletotrichum* sp.	leaf	Piracicaba, Brazil	[33]
	leaf, twig	Mirandela, Portugal	[27,29]
	fruit	Mirandela, Portugal	[28]
*Coniothyrium* sp.	leaf, twig	Sicily, Italy	[23]
*Coniozyma leucospermi*	leaf, twig	Mirandela, Portugal	[27,29]
*Coniozyma* sp.	flower buds	Bragança district, Portugal	[34]
*Cosmospora* sp.	leaf, twig	Mirandela, Portugal	[27,29]
*Cryptocoryneum* sp.	leaf	Salento, Italy	[21]
*Curvularia trifolii*	root	Bragança district, Portugal	[24]
*Cytospora pruinosa*	leaf, twig	Bragança district, Portugal	[24]
*Cytospora* sp.	stem, xylem	Majorca, Spain	[30]
	xylem	Apulia, Italy	[35]
*Dactylonectria pauciseptata*	root	Bragança district, Portugal	[24]
*Daldinia concentrica*	leaf	Piracicaba, Brazil	[33]
	branch	Ha’Ela Valley, Israel	[39]
*Dendrothyrium variisporum*	leaf, twig	Mirandela, Portugal	[27,29]
*Devriesia* sp.	branch, leaf	Salento, Italy	[21]
*Diaporthe ambigua*	root	Bragança district, Portugal	[24]
*Diaporthe columnaris*	leaf, root, twig	Bragança district, Portugal	[24]
*Diaporthe rudis*	twig	Mirandela, Portugal	[27]
*Diaporthe* sp.	stem	Majorca, Spain	[30]
	leaf	Sicily, Italy	[23]
	leaf	Piracicaba, Brazil	[33]
	leaf	Evora, Portugal	[31]
	root	Bragança district, Portugal	[24]
	flower buds	Bragança district, Portugal	[34]
	leaf	Alentejo, Portugal	[32]
	leaf, twig	Mirandela, Portugal	[27,29]
*Didymella macrostoma*	leaf	Alentejo, Portugal	[26]
*Didymella* sp.	branch	Salento, Italy	[21]
*Diplodia* sp.	leaf, twig	Sicily, Italy	[23]
*Discosia* sp.	leaf, twig	Mirandela, Portugal	[27,29]
*Dothiora oleae*	fruit	Karaburun, Turkey	(GenBank)
*Dothiorella iberica*	twig	Mirandela, Portugal	[27]
*Drechslera avenae*	leaf	Alentejo, Portugal	[26]
*Embellisia* sp.	leaf, twig	Mirandela, Portugal	[29]
*Endoconidioma populi*	leaf, twig	Mirandela, Portugal	[27,29]
*Epicoccum nigrum*	stem	Majorca, Spain	[30]
	leaf	Evora, Portugal	[31]
	root, twig	Bragança district, Portugal	[24]
	leaf	Alentejo, Portugal	[26]
	leaf, twig	Mirandela, Portugal	[27,29]
*Epicoccum* sp.	leaf	Sicily, Italy	[23]
	leaf	Alentejo, Portugal	[32]
*Eutypa tetragona*	leaf, twig	Mirandela, Portugal	[27,29]
*Eutypella* sp.	fruit	Bragança district, Portugal	[34]
*Exophiala* sp.	root	Córdoba, Spain	[22]
	branch	Salento, Italy	[21]
*Fimetariella rabenhorstii*	leaf, twig	Bragança district, Portugal	[24,27,29]
*Foliophoma* sp.	leaf	Alentejo, Portugal	[32]
*Fusarium lateritium*	fruit	Mirandela, Portugal	[28]
	leaf	Alentejo, Portugal	[26]
	twig	Mirandela, Portugal	[27]
*Fusarium musae*	leaf	Alentejo, Portugal	[26]
*Fusarium oxysporum*	root	Bragança district, Portugal	[24,40]
	twig	Mirandela, Portugal	[27]
*Fusarium* sp.	leaf	Evora, Portugal	[31]
	leaf, twig	Mirandela, Portugal	[27,29]
	leaf	Alentejo, Portugal	[32]
	xylem	Apulia, Italy	[35]
*Fusarium tricinctum*	leaf	Alentejo, Portugal	[26]
*Fusarium verticillioides*	leaf	Alentejo, Portugal	[26]
*Geopyxis* sp.	leaf, twig	Mirandela, Portugal	[29]
*Gibberella avenacea*	fruit	Mirandela, Portugal	[28]
*Gibberella* sp.	twig	Mirandela, Portugal	[27]
	branch	Salento, Italy	[21]
*Gloeosporium* sp.	leaf	Sicily, Italy	[23]
*Gloeotinia granigena*	leaf	Alentejo, Portugal	[26]
*Heydenia alpina*	twig	Mirandela, Portugal	[27]
*Heydenia* sp.	leaf, twig	Mirandela, Portugal	[27,29]
*Homortomyces* sp.	leaf, twig	Mirandela, Portugal	[29]
*Hormonema* sp.	stem	Majorca, Spain	[30]
	leaf	Salento, Italy	[21]
*Hortaea* sp.	branch, leaf	Salento, Italy	[21]
*Hyalodendriella betulae*	leaf, twig	Mirandela, Portugal	[27,29]
*Hyperphyscia* sp.	branch	Salento, Italy	[21]
*Hypoxylon* sp.	stem	Majorca, Spain	[30]
*Ilyonectria* sp.	leaf, twig	Mirandela, Portugal	[29]
*Kabatina* sp.	xylem	Majorca, Spain	[30]
*Lecania* sp.	branch, leaf	Salento, Italy	[21]
*Lecanora* sp.	branch, leaf	Salento, Italy	[21]
*Lecidella* sp.	branch, leaf	Salento, Italy	[21]
*Lecythophora* sp.	leaf, twig	Mirandela, Portugal	[29]
*Leimonis* sp.	branch	Salento, Italy	[21]
*Leptosphaerulina americana*	leaf	Alentejo, Portugal	[26]
*Leptosphaerulina australis*	leaf	Alentejo, Portugal	[26]
*Leptosphaerulina saccharicola*	leaf	Alentejo, Portugal	[26]
*Leptosphaerulina trifolii*	leaf	Alentejo, Portugal	[26]
*Libertasomyces platani*	xylem	Apulia, Italy	[35]
*Lophiostoma corticola*	root	Bragança district, Portugal	[24]
*Lophiostoma* sp.	branch, leaf	Salento, Italy	[21]
*Macrophomina phaseolina*	root	Bragança district, Portugal	[24]
*Macrophomina* sp.	root	Córdoba, Spain	[22]
*Microsphaeropsis arundinis*	root	Bragança district, Portugal	[24]
*Microsphaeropsis proteae*	leaf	Mirandela, Portugal	[27,29]
*Microsphaeropsis* sp.	stem	Majorca, Spain	[30]
	leaf, twig	Mirandela, Portugal	[27,29]
*Minimelanolocus* sp.	root	Córdoba, Spain	[22]
*Mycocalicium victoriae*	xylem	Apulia, Italy	[35]
*Mycosphaerella* sp.	leaf, twig	Mirandela, Portugal	[29]
*Naevala* sp.	leaf	Salento, Italy	[21]
*Nemania aenea*	leaf	Mirandela, Portugal	[29]
*Nemania* sp.	leaf, twig	Mirandela, Portugal	[29]
*Neocamarosporium* sp.	leaf, twig	Mirandela, Portugal	[29]
*Neocatenulostroma* sp.	branch, leaf	Salento, Italy	[21]
*Neocosmospora solani*	root	Siwa oasis, Egypt	[36]
	twig	Mirandela, Portugal	[27]
*Neodevriesia* sp.	branch, leaf	Salento, Italy	[21]
*Neofabraea kienholzii*	leaf, twig	Mirandela, Portugal	[27,29]
*Neofabraea* sp.	twig	Sicily, Italy	[23]
	fruit	Mirandela, Portugal	[28]
	leaf, twig	Mirandela, Portugal	[27,29]
	leaf	Alentejo, Portugal	[26]
*Neofabraea vagabunda*	leaf	Alentejo, Portugal	[26]
	fruit	Mirandela, Portugal	[28,34]
	leaf, twig	Mirandela, Portugal	[27]
*Neofusicoccum* sp.	leaf	Alentejo, Portugal	[32]
*Neophaeomoniella* sp.	branch, leaf	Salento, Italy	[21]
*Neosartorya* sp.	leaf	Alentejo, Portugal	[26]
*Neosetophoma* sp.	leaf	Salento, Italy	[21]
*Nigrospora oryzae*	stem	Majorca, Spain	[30]
	leaf	Evora, Portugal	[31]
*Nigrospora* sp.	leaf	Piracicaba, Brazil	[33]
	leaf	Alentejo, Portugal	[32]
*Ochrocladosporium* sp.	leaf, twig	Mirandela, Portugal	[29]
*Paecilomyces variotii*	xylem	Bisignano, Italy	[41]
*Paecilomyces verrucosus*	root	Bragança district, Portugal	[24]
*Paraconiothyrium* sp.	leaf, twig	Mirandela, Portugal	[29]
	xylem	Apulia, Italy	[35]
*Paraphaeosphaeria sporulosa*	root	Bragança district, Portugal	[24]
*Paraphoma chrysanthemicola*	leaf, root	Bragança district, Portugal	[24]
*Paraphoma* sp.	root	Bragança district, Portugal	[24]
*Parastagonospora avenae*	twig	Mirandela, Portugal	[27]
*Penicillium canescens*	root, twig	Bragança district, Portugal	[24]
	leaf	Trás-os-Montes, Portugal	[25]
*Penicillium chrysogenum*	leaf	Siwa oasis, Egypt	[42]
	fruit	Karaburun, Turkey	(GenBank)
*Penicillium citrinum*	fruit	Siwa oasis, Egypt	[36]
*Penicillium commune*	leaf, twig	Bragança district, Portugal	[24,43]
	leaf	Trás-os-Montes, Portugal	[25]
*Penicillium echinulatum*	leaf	Alentejo, Portugal	[26]
*Penicillium expansum*	leaf	Alentejo, Portugal	[26]
*Penicillium glabrum*	twig	Mirandela, Portugal	[27]
*Penicillium restrictum*	leaf, root	Bragança district, Portugal	[24]
*Penicillium roseopurpureum*	root	Bragança district, Portugal	[24,40]
*Penicillum* sp.	stem, xylem	Majorca, Spain	[30]
	leaf, twig	Sicily, Italy	[23]
	leaf, twig	Mirandela, Portugal	[27,29]
	xylem	Apulia, Italy	[35]
	branch, leaf	Salento, Italy	[21]
*Penicillium spinulosum*	leaf	Alentejo, Portugal	[26]
*Pestalotiopsis guepinii*	stem	Majorca, Spain	[30]
*Pestalotiopsis* sp.	leaf, twig	Mirandela, Portugal	[27,29]
*Phaeoacremonium* sp.	root	Córdoba, Spain	[22]
*Phaeococcomyces* sp.	branch, leaf	Salento, Italy	[21]
*Phaeohelotium* sp.	branch	Salento, Italy	[21]
*Phaeomoniella* sp.	leaf, twig	Mirandela, Portugal	[27,29]
	branch, leaf	Salento, Italy	[21]
*Phaeosphaeria* sp.	leaf, twig	Mirandela, Portugal	[27,29]
*Phaeothecoidea* sp.	branch, leaf	Salento, Italy	[21]
*Phoma herbarum*	leaf	Alentejo, Portugal	[26]
*Phoma* sp.	stem, xylem	Majorca, Spain	[30]
	leaf, twig	Sicily, Italy	[23]
	leaf, twig	Mirandela, Portugal	[27,29]
*Phyllosticta* sp.	leaf	Piracicaba, Brazil	[33]
*Pichia* sp.	branch	Salento, Italy	[21]
*Pithomyces chartarum*	xylem	Apulia, Italy	[35]
*Plectania rhytidia*	twig	Mirandela, Portugal	[27]
*Plectania* sp.	leaf, twig	Mirandela, Portugal	[29]
*Pleospora herbarum*	stem	Majorca, Spain	[30]
*Pleospora* sp.	leaf, twig	Sicily, Italy	[23]
*Pleurophoma* sp.	stem	Majorca, Spain	[30]
*Podospora* sp.	root	Bragança district, Portugal	[24]
*Preussia africana*	leaf	Alentejo, Portugal	[26]
*Preussia* sp.	leaf	Sicily, Italy	[23]
	shoot	Ljubljana, Slovenia	[37]
	leaf, twig	Mirandela, Portugal	[29]
	branch, leaf	Salento, Italy	[21]
*Prosthemium* sp.	leaf, twig	Mirandela, Portugal	[27,29]
*Pseudocamarosporium* sp.	xylem	Apulia, Italy	[35]
*Pseudocercospora* sp.	leaf, twig	Mirandela, Portugal	[27,29]
	branch, leaf	Salento, Italy	[21]
*Pseudocosmospora vilior*	root	Bragança district, Portugal	[24]
*Pseudophaeomoniella oleae*	xylem	Apulia, Italy	[35]
*Pseudophaeomoniella* sp.	leaf, twig	Mirandela, Portugal	[29]
	xylem	Apulia, Italy	[35]
*Purpureocillium lilacinum*	root	Bragança district, Portugal	[24,40]
*Purpureocillium* sp.	root	Córdoba, Spain	[22]
*Pycnidiophora* sp.	leaf, twig	Mirandela, Portugal	[29]
*Pyrenochaeta* sp.	leaf, twig	Mirandela, Portugal	[27,29]
*Pyronema domesticum*	leaf, twig	Mirandela, Portugal	[27,29]
*Pyrrhospora* sp.	branch, leaf	Salento, Italy	[21]
*Rachicladosporium* sp.	branch	Salento, Italy	[21]
*Ramularia* sp.	stem	Majorca, Spain	[30]
	branch, leaf	Salento, Italy	[21]
*Rhinocladiella similis*	leaf, twig	Mirandela, Portugal	[27,29]
*Rhinocladiella* sp.	branch, leaf	Salento, Italy	[21]
*Saccharata* sp.	leaf	Alentejo, Portugal	[32]
*Sarocladium* sp.	branch, leaf	Salento, Italy	[21]
*Scoliciosporum* sp.	branch, leaf	Salento, Italy	[21]
*Scutellinia* sp.	root	Córdoba, Spain	[22]
*Seimatosporium* sp.	leaf, twig	Mirandela, Portugal	[29]
*Seiridium* sp.	twig	Sicily, Italy	[23]
*Septoria* sp.	leaf, twig	Sicily, Italy	[23]
	leaf, twig	Mirandela, Portugal	[27,29]
*Sordaria macrospora*	stem, xylem	Majorca, Spain	[30]
	twig	Mirandela, Portugal	[27]
*Sordaria* sp.	leaf, twig	Mirandela, Portugal	[27,29]
*Sporormiella intermedia*	stem, xylem	Majorca, Spain	[30]
*Stagonosporopsis cucurbitacearum*	fruit	Karaburun, Turkey	(GenBank)
*Stemphylium solani*	leaf	Alentejo, Portugal	[26]
*Stemphylium* sp.	leaf, twig	Sicily, Italy	[23]
	branch, leaf	Salento, Italy	[21]
*Stemphylium vesicarium*	leaf	Alentejo, Portugal	[26]
	fruit	Karaburun, Turkey	[GenBank]
*Stigmatodiscus enigmaticus*	xylem	Apulia, Italy	[35]
*Talaromyces purpureogenus*	root	Portugal	[GenBank]
*Taphrina* sp.	branch, leaf	Salento, Italy	[21]
*Teratosphaeria* sp.	branch, leaf	Salento, Italy	[21]
*Tricharina* sp.	leaf, twig	Mirandela, Portugal	[27,29]
*Tricharina striispora*	twig	Mirandela, Portugal	[27]
*Trichoderma gamsii*	root, twig	Bragança district, Portugal	[24]
*Trichoderma koningii*	fruit	Mirandela, Portugal	[28]
*Trichoderma lixii*	root	Bragança district, Portugal	[24,40]
*Trichoderma polysporum*	stem	Majorca, Spain	[30]
*Trichoderma* sp.	root, twig	Bragança district, Portugal	[24]
	fruit	Mirandela, Portugal	[28]
	leaf, twig	Mirandela, Portugal	[29]
*Tumularia* sp.	leaf, twig	Mirandela, Portugal	[27,29]
*Valsa* sp.	leaf, twig	Mirandela, Portugal	[29]
*Valsaria* sp.	leaf, twig	Mirandela, Portugal	[29]
*Venturia* sp.	flower buds	Bragança district, Portugal	[34]
*Verticillium* sp.	leaf, twig	Mirandela, Portugal	[29]
*Wickerhamomyces* sp.	branch, leaf	Salento, Italy	[21]
*Xanthoparmelia* sp.	leaf	Salento, Italy	[21]
*Xanthoria* sp.	branch, leaf	Salento, Italy	[21]
*Xenosonderhenia* sp.	branch, leaf	Salento, Italy	[21]
*Xylaria* sp.	leaf	Piracicaba, Brazil	[33]
	twig	Mirandela, Portugal	[27]
*Zygoascus* sp.	branch	Salento, Italy	[21]

^1^ Species are reported according to the latest accepted name, which might not be the same as the one used in the corresponding reference.

**Table 2 microorganisms-08-01321-t002:** Other endophytic fungi reported from *Olea europaea*.

Endophyte ^1^	Plant Part	Country	Reference
Basidiomycota			
*Bullera* sp.	leaf	Alentejo, Portugal	[32]
*Chondrostereum purpureum*	fruit	Mirandela, Portugal	[28]
*Colacogloea* sp.	branch	Salento, Italy	[21]
*Conocybe* sp.	root	Córdoba, Spain	[22]
*Coprinellus* sp.	leaf, twig	Mirandela, Portugal	[27,29]
*Coriolopsis* sp.	fruit	Bragança district, Portugal	[34]
*Cryptococcus* sp.	leaf, twig	Mirandela, Portugal	[29]
	leaf	Alentejo, Portugal	[32]
	branch, leaf	Salento, Italy	[21]
*Cystofilobasidium* sp.	branch	Salento, Italy	[21]
*Dioszegia* sp.	branch	Salento, Italy	[21]
*Entoloma* sp.	root	Córdoba, Spain	[22]
*Erythrobasidium* sp.	leaf	Alentejo, Portugal	[32]
	branch, leaf	Salento, Italy	[21]
*Filobasidium* sp.	branch	Salento, Italy	[21]
*Kockovaella* sp.	branch, leaf	Salento, Italy	[21]
*Kondoa* sp.	branch	Salento, Italy	[21]
*Lepiota* sp.	root	Córdoba, Spain	[22]
*Malassezia* sp.	root	Córdoba, Spain	[22]
	branch, leaf	Salento, Italy	[21]
*Meira* sp.	branch	Salento, Italy	[21]
*Moniliophthora* sp.	root	Córdoba, Spain	[22]
*Peniophora cinerea*	leaf	Alentejo, Portugal	[26]
*Peniophora lycii*	leaf	Alentejo, Portugal	[26]
*Peniophora* sp.	leaf, twig	Mirandela, Portugal	[29]
	branch	Salento, Italy	[21]
*Phlebiopsis gigantea*	leaf	Alentejo, Portugal	[26]
*Porostereum* sp.	leaf, twig	Mirandela, Portugal	[29]
*Pseudomicrostroma* sp.	branch	Salento, Italy	[21]
*Quambalaria* sp.	branch, leaf	Salento, Italy	[21]
*Rhizoctonia* sp.	stem	Majorca, Spain	[30]
	leaf, twig	Sicily, Italy	[23]
*Rhodotorula mucilaginosa*	leaf	Alentejo, Portugal	[26]
*Sistotrema brinkmannii*	shoot	Ljubljana, Slovenia	[37]
*Sporobolomyces* sp.	leaf	Alentejo, Portugal	[32]
	branch, leaf	Salento, Italy	[21]
*Sporotrichum* sp.	leaf	Sicily, Italy	[23]
*Symmetrospora* sp.	branch, leaf	Salento, Italy	[21]
*Trametes* sp.	leaf, twig	Mirandela, Portugal	[27,29]
	branch, leaf	Salento, Italy	[21]
*Tremella* sp.	branch	Salento, Italy	[21]
*Tricholoma* sp.	leaf, twig	Mirandela, Portugal	[29]
*Vishniacozyma* sp.	branch, leaf	Salento, Italy	[21]
*Wallemia* sp.	leaf	Salento, Italy	[21]
*Xylobolus annosus*	twig	Mirandela, Portugal	[27]
Mucoromycota			
*Mucor plumbeus*	stem, xylem	Majorca, Spain	[30]
*Mucor racemosus*	stem	Majorca, Spain	[30]
*Rhizopus arrhizus*	root	Bragança district, Portugal	[24]
*Rhizopus* sp.	leaf	Alentejo, Portugal	[26]
*Rhizopus stolonifer*	fruit	Karaburun, Turkey	(GenBank)
*Umbelopsis vinacea*	stem	Majorca, Spain	[30]

^1^ Species are reported according to the latest accepted name, which might not be the same as the one used in the corresponding reference.

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
