# Peer review of "Endophytic Fungi of Olive Tree"

_microorganisms, 2020, doi:10.3390/microorganisms8091321_

Round 1

Reviewer 1 Report

Dear authors,

The review, Endophytic Fungi of Olive Tree is well written review and the topic is relevant and interesting.  Authors need to pay attention to certain points in the review.  Authors should consider the comments useful for further revision of the manuscript.

Comments:

Lines (212 and 260)- Authors mention about the defensive mutualism. What do authors mean by this term ?

Line 223 and 75- Authors talk about plant hormones. It is important to include examples for those plant hormones.

Authors should consider adding the following points as they are important in the context of endophytic fungi

What are the influences of ecological environments and genetic background/tissues of Olive tree on the population structure of endophytic fungi?

Endophytic fungi could enhance the resistance of host plants to biotic and abiotic stresses by producing bioactive compounds (chemicals). What are some examples in the case of endophytic fungi of Olive tree?

What are the potential promotion of fitness and growth of Olive trees because of endophytic fungi ?

What are the possible examples of biocontrol by Trichoderma

Author Response

Response to Reviewer 1 Comments

R1: The review, Endophytic Fungi of Olive Tree is well written review and the topic is relevant and interesting.  Authors need to pay attention to certain points in the review.  Authors should consider the comments useful for further revision of the manuscript.

AU1: We really appreciate an thank the Reviewer 1 for the positive evaluation of the review and also the precious suggestions that surely allow us to improve the manuscript

R2: Lines (212 and 260)- Authors mention about the defensive mutualism. What do authors mean by this term?

AU2: An explanation of the concept of ‘defensive mutualism’ is provided in a few cited references, such as [12,15,16,19], which have been added at line 212.

R3: Line 223 and 75- Authors talk about plant hormones. It is important to include examples for those plant hormones.

AU3: Line 223 has been modified as follows:  ‘… such as auxins by Discosia sp. [89], and gibberellins by Phoma herbarum [90]. Moreover, several yeast species are known for these effects, such as Aureobasidium pullulans which stands out for its widespread occurrence on the olive phylloplane too [41,91]’.

R4: Authors should consider adding the following points as they are important in the context of endophytic fungi: What are the influences of ecological environments and genetic background/tissues of Olive tree on the population structure of endophytic fungi?

AU4: In this respect, the available information is very limited (see lines 107-109), and nothing can be inferred until additional data are obtained. Actually, one of the scope of this review is to stimulate further investigations in the field to improve the current knowledge.

R5: Endophytic fungi could enhance the resistance of host plants to biotic and abiotic stresses by producing bioactive compounds (chemicals). What are some examples in the case of endophytic fungi of Olive tree?

AU5: All the examples currently available from the literature (actually quite few) are reported in section 4.

R6: What are the potential promotion of fitness and growth of Olive trees because of endophytic fungi?

AU6: This is outlined in section 3.2.

R7: What are the possible examples of biocontrol by Trichoderma

AU7: Lines 195-197 have been modified as follows: ‘…species,  have already been experimentally evaluated on olive tree, with reference to both kinds of beneficial effects, particularly for the biocontrol of V. dahliae and N. solani [18,69-71].’

Reviewer 2 Report

This is an interesting overview of the current state of endophyte research in olive tree species.  My main concern is int he language use (grammar and sentence structure).  This paper would be greatly improved by another editing procedure where more common phrasing and sentence structure are used.  This should be published but , in my opinion, needs a detailed editing.  I would hate to see this not available to the field and look forward to reading it once published.

Author Response

Response to Reviewer 2 Comments

R1: This is an interesting overview of the current state of endophyte research in olive tree species.  I would hate to see this not available to the field and look forward to reading it once published.

AU1: We thank the Reviewer 2 for the enthusiastic evaluation of the review.

R2: My main concern is int he language use (grammar and sentence structure).  This paper would be greatly improved by another editing procedure where more common phrasing and sentence structure are used.  This should be published but , in my opinion, needs a detailed editing.

AU2: With reference to comment by rev. #2, we want to refer that the text was already reviewed and undergone to an internal English editing by an English mother tongue colleague of ours before the submission, to avoid use of ‘common phrasing and sentence structure’.

In the absence of specific remarks by this referee concerning grammar mistakes or other aspects to be corrected, and considering the positive judgment by rev. #1 (five stars for correct and readable English), we did not consider to bring further modifications to it.
